# Systematic Review of the Survival Outcomes of Neoadjuvant Chemotherapy in Women with Malignant Ovarian Germ Cell Tumors

**DOI:** 10.3390/cancers15184470

**Published:** 2023-09-08

**Authors:** Hitomi Sakaguchi-Mukaida, Shinya Matsuzaki, Yutaka Ueda, Satoko Matsuzaki, Mamoru Kakuda, Misooja Lee, Satoki Deguchi, Mina Sakata, Michihide Maeda, Reisa Kakubari, Tsuyoshi Hisa, Seiji Mabuchi, Shoji Kamiura

**Affiliations:** 1Department of Gynecology, Osaka International Cancer Institute, Osaka 540-0008, Japan; 2Department of Obstetrics and Gynecology, Graduate School of Medicine, Osaka University, Osaka 565-0871, Japan; 3Department of Obstetrics and Gynecology, Osaka General Medical Center, Osaka 558-8558, Japan; 4Department of Forensic Medicine, School of Medicine, Kindai University, Osaka 577-8502, Japan

**Keywords:** germ cell, ovarian cancer, malignant germ cell tumor, neoadjuvant chemotherapy, advanced disease, systematic review

## Abstract

**Simple Summary:**

This systematic review was conducted using four public electronic databases (PubMed, Web of Science, Scopus, and the Cochrane Central Register of Controlled Trials) from the date of inception to 31 May 2023, and 10 original articles with information regarding neoadjuvant chemotherapy (NACT) for malignant ovarian germ cell tumors (MOGCT) were identified. The results of the meta-analysis showed that NACT was used in approximately 40% of advanced MOGCT cases, with a response rate of 95.8%. The uterine preservation rate was approximately 70%, and the cumulative resumed menstruation rate was 100% (*n* = 30). Four comparator studies evaluating NACT versus primary debulking surgery between the groups showed similar overall survival, disease-free survival, recurrence rate, and adverse event rate (grade 3–4) from chemotherapy. Although available data are limited and level I evidence is lacking, we believe that NACT may be feasible for advanced MOGCT. Further studies are required to confirm this study’s results.

**Abstract:**

Randomized clinical trials assessing the efficacy of neoadjuvant chemotherapy (NACT) for advanced epithelial ovarian cancer have predominantly included women with high-grade serous carcinomas. The response rate and oncological outcomes of NACT for malignant ovarian germ cell tumors (MOGCT) are poorly understood. This study aimed to examine the effects of NACT on women with MOGCT by conducting a systematic review of four public search engines. Fifteen studies were identified, and a further descriptive analysis was performed for 10 original articles. In those studies, most women were treated with a bleomycin, etoposide, and cisplatin regimen, and one to three cycles were used in most studies. Four studies comparing NACT and primary debulking surgery showed similar complete response rates (*n* = 2; pooled odds ratio [OR] 0.90, 95% confidence interval [CI] 0.15–5.27), comparable overall survival (*n* = 3; 87.0–100% versus 70.0–100%), disease-free survival (*n* = 3; 87.0–100% versus 70.0–100%), recurrence rate (*n* = 1; OR 3.50, 95%CI 0.38–32.50), and adverse events rate from chemotherapy between the groups. In conclusion, NACT may be considered for the management of MOGCT; however, possible candidates for NACT use and an ideal number of NACT cycles remain unknown. Further studies are warranted to validate the efficacy of NACT in advanced MOGCT patients.

## 1. Introduction

In 2023, ovarian cancer will become the fifth most lethal malignant disease among women in the United States (U.S.) [1]. Most ovarian cancers are epithelial, such as high-grade serous carcinoma (HGSC), endometrioid, clear-cell carcinoma (CCC), and mucinous carcinoma (MOC). Non-epithelial ovarian tumors account for approximately 10% of ovarian cancer cases, representing approximately 2200 new cases per year in the U.S., and the most common subtype of non-epithelial ovarian tumors arising from germ cells is malignant ovarian germ cell tumor (MOGCT) [2]. MOGCTs are typically unilateral ovarian tumors occurring in young women aged 18–30 years and are characterized by high chemosensitivity and rapid tumor growth [3,4]. MOGCT is an aggressive type of malignant disease, and some cases show advanced disease, such as numerous abdominal metastases, a large tumor size, and massive pleural effusion. Primary debulking surgery (PDS) for treating advanced MOGCT may be associated with high surgical morbidity [5,6,7]. Generally, MOGCT is treated with PDS, followed by chemotherapy, and most women with these types of tumors are cured due to the high chemosensitivity of MOGCT [8]. Nevertheless, PDS is often challenging for women with advanced MOGCT [9].

Four randomized controlled trials (RCT) examining the effectiveness of neoadjuvant chemotherapy (NACT) for treating advanced epithelial ovarian cancer (EOC) predominantly included patients with HGSC [10,11,12,13]. In these RCTs, NACT followed by interval debulking surgery (IDS) was found to be non-inferior to PDS followed by chemotherapy as a treatment option for women with advanced ovarian cancer. Therefore, NACT is the preferred treatment option in advanced EOC for preventing surgical morbidity [14,15,16,17]. Nevertheless, these RCTs did not include women with MOGCT, and no other RCTs on this topic are available; this may be due to the rarity of MOGCT. Notably, a systematic review focusing on the efficacy of NACT in women with MOGCT has not been published. Therefore, in contrast to EOC, the effect of NACT on women with MOGCT is understudied.

We hypothesized that since MOGCT is highly chemosensitive, NACT followed by IDS may be a useful treatment modality for advanced MOGCT cases. This systematic review aimed to examine the effects of NACT on women with MOGCT.

## 2. Materials and Methods

### 2.1. Ethics Statement

As the current systematic review used publicly available and de-identified data, the Institutional Review Board of Osaka International Cancer Institution has exempted our present study from the requirements of informed patient consent. Any unpublished patient data were excluded from this study.

### 2.2. Approach to a Systematic Literature Review

This systematic review was registered with the International Prospective Register of Systematic Reviews (Registration ID: CRD42023434946). First, a systematic search of the literature regarding MOGCT using keywords related to Medical Subject Headings (MeSH) and non-MeSH terms was conducted (Appendix A). Thereafter, studies examining the effects of NACT on women with MOGCT were selected. The primary objective of the screened studies was to determine the response rate to NACT in women with MOGCT. The secondary objective was to explore the disease-free survival of women with MOGCT treated with NACT. Additional outcomes included surgical outcomes of IDS after NACT in women with MOGCT, rate of menstruation resumption, and adverse effects of chemotherapy.

### 2.3. Eligibility Criteria, Information Sources, and Search Strategy

Four publicly accessible electronic databases (PubMed, Web of Science, Scopus, and the Cochrane Central Register of Controlled Trials) were used to perform a systematic literature search from the inception of these databases to 31 May 2023, according to the Preferred Reporting Items for Systematic Reviews and Meta-Analyses guidelines [18]. The search keywords used to identify relevant studies are listed in Appendix A. MeSH terms were searched in the PubMed and Cochrane databases to identify articles on NACT use in women with MOGCT, as previously described [19,20].

Published articles on NACT use for MOGCT were screened by checking the titles, abstracts, and full texts of candidate published studies, as previously described, with some modifications [21,22]. All titles and abstracts were screened by H.S. and S.M.

### 2.4. Study Selection

The inclusion criteria of the current study were as follows: (1) comparative studies performed between an experimental group (such as NACT followed by IDS and NACT only) and a control group (PDS followed by adjuvant chemotherapy); (2) studies providing sufficient information regarding the survival outcomes of women with MOGCT who received NACT; (3) studies containing sufficient information to investigate relevant outcomes; and (4) studies with clear information about the number of women with MOGCT.

Herein, the following exclusion criteria were used: (1) insufficient information to identify the prognosis of patients; (2) language other than English; (3) reviews, systematic reviews, and meta-analyses; and (4) gray literature such as conference abstracts, conference papers, government reports, committee reports, and ongoing research [23].

### 2.5. Data Extraction

The data were independently extracted by H.S.-M. and Sh.M. using Excel 2021 (Microsoft) and cross-checked against each other. The following information was recorded: publication year, study location, name of the lead author, number of included cases, number of cases where NACT was used, and outcomes of interest.

### 2.6. Analysis of Outcome Measures and Assessment of Bias Risk

The outcomes of interest in this study were the rate of NACT use, NACT response rate, NACT regimen, number of NACT cycles, survival outcomes (overall survival [OS] and disease-free survival [DFS]), surgical outcomes of IDS after NACT, and reproductive outcomes after NACT use. Patient-level analysis was also performed by combining information from case reports and case series.

As in our previous studies, the Risk of Bias in Non-randomized Studies-of Interventions (ROBINS-I) tool was used to assess the risk of bias in eligible studies [24,25,26].

### 2.7. Sensitivity Analysis

In the sensitivity analysis, the response rate and survival outcomes were examined according to the histological type of MOGCT.

### 2.8. Meta-Analysis Plan

The odds ratios (ORs) of the risk of outcomes (response rate and rate of adverse events) and hazard ratios of survival outcomes extracted from the retrieved studies were calculated using the 95% confidence intervals (95%CI) of the reported values. To calculate the rate of NACT use, the number of women with MOGCT treated with NACT was divided by the total number of women included in the MOGCT cohort.

Heterogeneity among studies was assessed using the *I*^2^ statistic, which measures the percentage of total variation among the included studies. Heterogeneity was assessed based on the value of *I*^2^ (low [0–30%]; moderate [30–60%]; substantial [50–90%]; and considerable [75–100%]), with some modifications from that reported in a previous study and as per the Cochrane Handbook for Systematic Reviews of Interventions, version 6.3 [27]. In the pooled analysis, a fixed-effects analysis was performed for low heterogeneity, and a random-effects analysis was adopted for moderate or greater heterogeneity.

RevMan, version 5.4.1 software (Cochrane Collaboration, Copenhagen, Denmark), was used to conduct meta-analyses. Data for all outcomes were entered into RevMan 5.4.1 so that negative effect sizes or relative risks of less than one favored active intervention.

### 2.9. Statistical Analysis

To examine differences in the baseline demographics between the two groups (i.e., NACT versus the non-NACT group), Fisher’s exact test or the Chi-squared test were used as appropriate. All statistical analyses were based on two-tailed hypotheses, and statistical significance was set at *p* < 0.05. SPSS version 28.0 (IBM Corp., Armonk, NY, USA) was used in the analyses.

## 3. Results

### 3.1. Results of the Systematic Review

#### 3.1.1. Study Selection

The study selection was performed as shown in Figure 1. A total of 141 articles were screened, and 15 studies, comprising 142 women with MOGCT treated with NACT versus 387 women with MOGCT treated without NACT, met the inclusion criteria for the present systematic review [5,6,7,9,28,29,30,31,32,33,34,35,36,37,38]. After the identification of eligible studies, we checked the reference lists of all eligible studies and confirmed that potential citations/studies were not missed during the literature screening.

After excluding duplicate studies, 141 potentially eligible studies were screened. First, 77 studies were excluded after a title review. Second, 30 studies were excluded during an abstract review. Third, 19 studies were excluded after a full-text review; reasons for exclusion are presented in Figure 1.

#### 3.1.2. Study Characteristics

In Appendix A, the metadata from the 15 included studies is summarized. All of the included studies (*n* = 15) were retrospective in nature and were published between 1999 and 2023. Of these 15 studies, 10 were original articles [9,28,29,30,31,32,33,34,36,37] and five were case reports [5,6,7,35,38]. Approximately half of the studies (*n* = 8) were conducted in India [6,28,30,31,32,34,35,36], followed by China (*n* = 2) [9,33] and others (*n* = 1: England, France, Spain, Thailand, and Japan) [5,7,29,37,38]. Four comparator studies compared NACT versus non-NACT use in women with MOGCT [9,30,32,34], and the remaining 11 were non-comparator studies. Because case reports did not add new findings to the current systematic review, a descriptive analysis was not performed.

#### 3.1.3. Risk of Bias in Included Studies

The four comparator studies included in the present review were assessed using ROBINS-I and demonstrated possible severe bias (low quality) in all studies (Appendix A) [9,30,32,34]. Because case-report analysis was not performed, the risk of bias was not assessed for case reports.

### 3.2. Results of the Meta-Analysis

#### 3.2.1. Overview of Included Studies

Among the 15 studies included in this review, the median number of included cases and NACT use cases was 31 (range, 1–138) and four (range, 2–27), respectively (Table 1). The histological types of MOGCT included in this review are listed in Appendix A. The indication for NACT use was clarified in 10 studies: NACT was used due to advanced disease in eight studies and advanced disease with poor performance status in two studies. The median age was available in six studies and ranged from 17 to 23.6 years (Table 1). The number of women with stages II–IV MOGCT was clarified in six studies, and the rate ranged from 35.4% to 100%.

#### 3.2.2. NACT Rate, Regimen, and Cycles

Among the original articles (*n* = 10), the rate of NACT use in women with MOGCT ranged from 6.5 to 69.2% in the whole cohort (number of NACT use/number of women with all stages) and from 28.6 to 81.8% in those with advanced disease (NACT use/women with stages II–IV). The cumulative rate of NACT use was 136 of 525 (25.9%) patients in the entire cohort and 110 of 266 (41.4%) patients with advanced disease.

Among the 136 women administered NACT in the 10 original articles, the regimen was clarified for 76 patients. Most women (96.0%) were treated with Bleomycin, Etoposide, and Cisplatin (BEP), followed by Bleomycin, Etoposide, and Carboplatin (2.6%), Paclitaxel and Carboplatin (1.3%), and Cisplatin with Cyclophosphamide (1.3%). Among the 10 studies, NACT regimen cycles were clarified in six studies, as shown in Table 1. Four of the six studies included >10 women who underwent NACT. Of those studies (*n* = 4), two used one to three cycles, and two used four cycles of the NACT regimen.

#### 3.2.3. Comparator Studies

Four studies compared the outcomes of interest, as defined in our present study, between NACT and non-NACT groups (Table 2) [9,30,32,34]. The number of studies comparing relevant outcomes, enumerated in our present review, are as follows: response rate (*n* = 2) [30,34], survival rate (*n* = 3) [9,30,34], survival rate in large tumors (>20 cm) (*n* = 1) [9], recurrence rate (*n* = 1) [32], surgical outcome of PDS (*n* = 1) [9], the rate of resumed menstruation (*n* = 2) [30,34], and adverse effect of chemotherapy (*n* = 1) [34].

##### Primary Outcomes: Response Rate

Two studies clarified the response rate to chemotherapy in the NACT and non-NACT groups [30,34]. A similar complete response (CR) rate was observed in the studies by Talukdar et al. (OR 1.49, 95% confidence interval (CI) 0.51–4.39) and Agarwal et al. (OR 0.19, 95%CI 0.01–4.29) (Figure 2). Pooled OR using random-effect analysis has shown similar CR rates between NACT and non-NACT groups (OR 0.90, 95%CI 0.15–5.27; heterogeneity: *p* = 0.21, *I*^2^ = 35%).

The pooled odds ratios were calculated using random-effects analysis for the unadjusted analysis of the response rate to chemotherapy for women with NACT and PDS. Moderate heterogeneity was observed during the unadjusted analysis (*I*^2^ = 35%). Abbreviations: MOGCT, malignant ovarian germ cell tumor; NACT, neoadjuvant chemotherapy; PDS, primary debulking surgery.

In the study by Talukdar et al., the response rate (CR + partial response [PR]) was available, and similar values of response rates were observed in the two groups (OR 3.18, 95%CI 0.63–15.98) [34]. No studies have examined the response rate according to the histological subtypes of MOGCT (sensitivity analysis).

##### Co-Primary and Secondary Outcomes: Survival Outcomes

The 5-yr OS and DFS rates were reported in three studies [9,30,34]. Two studies concluded that DFS was similar between the NACT and PDS groups (100% versus 100% in Agarwal et al.’s study and 95.3% versus 96.9% in Lu et al.’s study), while two studies showed almost identical OS values (100% versus 100% in Agarwal et al.’s study and 95.3% versus 96.9% in the study by Lu et al.) between the two groups. In the study by Talukdar et al., the 5-yr OS and DFS rates were 87.0% in the NACT group and 69.8% in the PDS group; however, a statistical analysis was not performed to compare the survival outcomes between the two groups [34].

One study performed a sensitivity analysis, comparing the 3-yr DFS rate of large tumors (>20 cm) between the NACT (*n* = 16, 80.0%) and PDS (*n* = 4, 25.0%) groups [9]. While the 3-yr DFS rate appeared to be better in the NACT group, a statistical analysis was not performed.

##### Additional Outcomes

The results of surgery between NACT and PDS groups were determined in the study by Lu et al. [9]. In this study, the rate of optimal surgery (residual disease ≤ 2 cm) was 89%, and the rate was similar between the two groups (NACT: 95% versus PDS: 84%, *p* = 0.384), whereas women with NACT were more likely to undergo R0 resection compared to those with PDS (NACT: 81% versus PDS: 44%, *p* = 0.007). The median intraoperative blood loss was significantly lower in the NACT group compared to the PDS group (NACT: 200 versus PDS: 450 mL, *p* = 0.018). The remaining surgical outcomes and postoperative complications were similar between the two groups (Table 2). The rates of adverse effects of chemotherapy were examined in one retrospective study, and the rates of adverse effects (grades 3 and 4) were similar between the NACT and PDS groups [9,34]. In specific, the rates of leukopenia (6/23 [26.8%] versus 11/43 [25.5%]) and thrombocytopenia (1/23 [4.3%] versus 6/43 [14.0%]) were similar between the NACT and PDS groups. No pulmonary toxicity due to bleomycin was observed in both the NACT and PDS groups [34].

The rate of resumed menstruation was determined in two studies [30,34]. In the study by Agarwal et al., all women in the NACT and PDS groups resumed menstruation (100% versus 100%; OR, not estimated). In the study by Talukdar et al., women with NACT use were more likely to have a higher rate of resumed menstruation (OR 4.59, 95% CI 1.11–18.98) compared to those without NACT use.

#### 3.2.4. Non-Comparator Studies

Ten retrospective studies were included in the present systematic review [9,28,29,30,31,32,33,34,36,37] and six non-comparator studies had sufficient information regarding the outcome of interest (Table 3) [9,30,32,33,34,36]. In this section, as no studies other than the four comparator studies were identified for each outcome, the outcomes are not mentioned. Several studies have compared the relevant outcomes enumerated in the present review: response rate to NACT (*n* = 6) [9,30,32,33,34,36], survival rate (*n* = 4) [9,30,33,34], and rates of menstruation and pregnancy resumption (*n* = 4) [28,30,33,34].

##### Primary Outcomes: Response Rate 

Six studies, including 104 women who underwent NACT, were included in this analysis [9,30,32,33,34,36]. All studies (*n* = 6), including 104 NACT patients, clarified the number of CRs, and the cumulative CR rate was 31.7% (33/104 women). With regard to the response (CR + PR) for NACT, four studies, including 72 MOGCT patients who underwent NACT, clarified the number of both CR and PR, and the response rate to NACT (CR + PR) was 95.8% (69/72 women). Among the 104 women who underwent NACT, one (1.0%) showed disease progression during NACT.

##### Co-Primary and Secondary Outcomes: Survival Outcomes

Four studies (three comparators and one non-comparator study) have shown the survival outcomes from the use of NACT for women with MOGCT (Table 4) [9,30,33,34] and the data from three studies have already been shown in Section 4.3.2 [9,30,34]. One non-comparator study, including 18 women who underwent NACT for yolk sac tumor (YST), showed that the 5-yr DFS and OS rates were both 94.4% [33].

##### Additional Outcomes: Rates of Hysterectomy, Resumed Menstruation, and Pregnancy

Based on the available information from five studies with 79 MOGCT patients who underwent NACT, a hysterectomy was performed in 27 women (34.2%) (Table 5) [28,30,32,33,34]. In other words, the cumulative rate of fertility-sparing surgery was 66.8%. This implies that the rate of fertility-sparing surgery ranged from 18.5% to 100% among the studies included in this review. However, information regarding the uterine preservation rate and the desire to preserve fertility was unavailable. The rate of menstruation resumption was determined in three studies [28,30,34], and the pregnancy rate was examined in two studies [33,34]. Among the women who did not undergo hysterectomy, the cumulative rate of resumed menstruation was 100% (30/30), and the cumulative pregnancy rate was 100% (16/16) (Table 5). No adverse events during pregnancy were reported in any of the included studies.

## 4. Discussion

### 4.1. Principal Findings

The principal findings of this systematic review are as follows: (*i*) the response rate to NACT was approximately 95% and considered to be high; (*ii*) the survival outcomes appeared to be comparable between the NACT and PDS groups; (*iii*) NACT has the potential to improve the survival outcomes of advanced MOGCT, such as large tumors (>20 cm); and (*iv*) the rate of resumed menstruation was 100%. As no systematic review has focused on the use of NACT for MOGCT, we believe that the results of the current systematic review are the first to appear in the literature. Nevertheless, MOGCT is a rare disease and has resulted in the retrieval of only a limited number of studies. Therefore, the results of the present review require careful interpretation.

### 4.2. Strengths and Limitations

This systematic review has two strengths. To the best of our knowledge, this is the first systematic review focusing on the outcomes of women with MOGCT who were treated with NACT. Although the available data are limited and the outcomes cannot be determined according to the histologic type of MOGCT, NACT use appears to be feasible both in terms of survival outcomes and adverse events. Nevertheless, to confirm our findings, the use of NACT in MOGCT cases should be determined in future studies. Since the included studies focused on NACT use, the results of this review may have a strong bias; thus, a large-scale study, such as a multi-institutional or nationwide study, is warranted to resolve this problem. In the past, we have successfully examined the outcomes of the less common histological subtypes of ovarian cancer (CCC, MOC, and low-grade serous carcinoma [LGSC]) using the Surveillance, Epidemiology, and End Results (SEER) and NCDB databases [39]. We believe that these previously tested approaches may be useful for analyzing the outcomes of NACT in MOGCT in the future.

Nevertheless, this study has several limitations. First, since all identified studies were retrospective in nature, unmeasured bias (such as unmatched patient background, a limited number of women with MOGCT, and most studies having been reported in one country (India])) may exist. Second, the current systematic review had a limited number of included cases; thus, no studies were able to exclude confounding factors using multivariate analysis or propensity matching analysis. Moreover, all studies did not use advanced statistical methods. For instance, all comparator studies comparing NACT versus PDS (*n* = 4) did not calculate the hazard ratio for OS and DFS. Thus, the pooled hazard ratio could not be calculated in the current study. Third, the decision to use NACT lies at the discretion of the clinician, which may lead to severe healthcare provider bias. For instance, advanced inoperable cases or women with poor performance status may be selected for NACT, and this may lead to poor outcomes for NACT use in MOGCT cases. Thus, an RCT or a patient-background-matched study is necessary in the future. Fourth, the current study may have had a publication bias. For instance, studies on women with a poor prognosis after NACT use may not have been published. This publication bias may not be eliminated in the analysis and needs to be recognized as a key drawback with cautious interpretation. Fifth, MOGCT is a rare type of ovarian cancer, and no studies have assessed its central pathology. Therefore, the accuracy of the MOGCT diagnosis remains unknown. While chemosensitivity and survival outcomes differ among dysgerminomas, YST, and immature teratomas, the current systematic review does not have enough information to examine the outcome of interest according to the histological subtypes of MOGCT [40,41,42,43,44]. Sixth, the meta-analysis may have limited value owing to the small number of published articles. Because of the limited data available, only limited information on patient background could be presented. Finally, most studies were conducted in India and China, and generalizability to other regions has not been evaluated. Notably, the majority of studies (*n* = 8) originated from India, negatively impacting the generalizability of the data.

### 4.3. Comparison with Existing Literature

A nationwide U.S. study using the SEER database between 1994 and 2014 included 2238 women with MOGCT and evaluated the surgical management and prognostic factors of MOGCT [45]. In this nationwide study, approximately one-third of the women had advanced disease (stages II–IV: *n* = 657 [30.7%]), and hysterectomy was performed in approximately 10% of the women (stages I–IV: *n* = 234 [12.4%]). In another U.S. population-based study using the National Cancer Database, the rate of advanced MOGCT (stages II–IV) was 34.5%, and hysterectomy was performed in approximately 20% of patients with advanced MOGCT [46]. These data suggest that advanced MOGCT is rare and that women with possible NACT are also rare. The rarity of this disease is a barrier to elucidating the efficacy of NACT.

With regard to the treatment of advanced MOGCT, most national guidelines suggest PDS with adjuvant chemotherapy [8,47,48,49]. For instance, in the National Comprehensive Cancer Network (NCCN) guidelines, NACT refers to the treatment administered to reduce the tumor burden before debulking surgery for advanced EOC (stages II–IV) [8]. On the other hand, initial surgery followed by adjuvant chemotherapy is suggested for advanced MOGCT cases, and NACT is not listed as a treatment option. Conversely, NACT is recommended for women with advanced MOGCT (stage IC/IIA and greater) by the Royal College of Obstetricians and Gynecology (RCOG) Green Top Guidelines [50]. These policy discrepancies are due to the lack of level I evidence and rarity of the disease; thus, further research is warranted to demonstrate the efficacy of NACT for advanced MOGCT. The RCOG recommends NACT use for women with stage IC/IIA MOGCT, and the examination of NACT candidates is also warranted.

#### 4.3.1. Primary Outcomes: Response Rate

Chemosensitivity is an important factor in NACT for EOC, and HGSC shows a good response (65–80%) to platinum-based chemotherapy [51,52]. Nevertheless, CCC, MOC, and LGSC are considered less chemoresponsive (10–30%) than conventional platinum-based chemotherapy [53,54,55,56,57,58,59,60,61]. Differences in chemosensitivity based on the histological type of the tumor may result in different outcomes stemming from NACT for EOC. The results of recent population-based studies using SEER or NCDB showed that the efficacy of NACT appeared to be better in HGSC than in those with less chemosensitive histologic subtypes (CCC, MOC, and LGSC) [39,62].

The response rate to NACT may be important to evaluate the chemosensitivity of MOGCTs. In the present systematic review, the rate of CR was 31.7%, and the response rate (CR + PR) was 95.8%; these results are considered high and chemosensitive. We believe that NACT is a good treatment option for MOGCT. Nevertheless, the lack of data based on the histological subtypes of MOGCT is a strong limitation of the current review, and further research is warranted.

The evaluation of the response to NACT is important, and the timing and method of determination are essential; however, the timing of evaluation was discussed in only one study. In the study reported by Zhang et al. [33], one to two cycles of the BEP regimen were considered sufficient to achieve a complete or nearly CR. Thus, they propose a re-evaluation within two cycles of NACT. The method of evaluation was not discussed in all studies. In our estimation, computed tomography may be used to evaluate the response to NACT.

While CA125 has been determined as a possible predictor of complete resection during IDS after NACT in EOC [63,64,65], no studies have evaluated MOGCT serum markers to predict the response to NACT or the feasibility of IDS. In the study reported by Lu et al. [9], a significant decrease in median serum AFP level was observed after NACT (3.3 × 10^4^ to 6.6 × 10^2^ ng/mL, *p* < 0.001). We expect that AFP is a potential predictor of the response to NACT and the feasibility of IDS.

A poor response to NACT was observed in 3 out of 72 patients. Among these patients (*n* = 3), 2 had stable disease (SD) and 1 had progression disease (PD); the clinical course of 2 women (SD 1, PD 1) was clarified. In a woman with PD, NACT was discontinued and IDS was performed after one course of NACT; the survival outcome was unavailable. In a woman with SD, NACT was discontinued owing to the elevated serum AFP level, and there was no significant response to NACT based on radiological findings. IDS was performed after one course of NACT and cytoreduced to macroscopic residual disease of 2 cm or less. Contrary to what was ascertained in the radiologic findings, most of the tumor was necrotic, and IDS was performed without severe complications. Poor response to NACT may lead to poor outcomes; thus, prompt discontinuation of NACT and maximal effort in IDS were performed in the previous reports.

#### 4.3.2. Co-Primary and Secondary Outcomes: OS and DFS

The oncologic outcomes of women with MOGCT who were treated with NACT appeared to be similar to those treated with PDS; however, the number of NACT cases was small (*n* = 11–23), and type II errors need to be recognized. Moreover, the patient backgrounds and tumor characteristics were not matched using multivariate analysis, propensity score matching, or inverse probability of treatment weight analysis because of the insufficient number of eligible cases. An RCT is needed to examine the effects of NACT on MOGCT; however, this may be difficult because of the rarity of MOGCT cases. To resolve this matter, a nationwide study that matches the patient background and tumor characteristics of women with MOGCT comparing PDS versus NACT may be useful. We recently published a nationwide study that examined the effects of NACT in women with the less common EOC. This approach may be adapted for studies examining the effect of NACT on MOGCT [39].

Possible candidates for NACT use and the ideal number of chemotherapy cycles are understudied. A Chinese retrospective study suggested that NACT followed by IDS has the potential to show superior DFS for large tumors (>20 cm) compared to those treated with PDS [9]. The study also found that women with MOGCT who were treated with NACT were more likely to have a higher rate of R0 resection (81% versus 44%) and less intraoperative blood loss (200 mL versus 450 mL) compared to those treated with PDS. Therefore, advanced-stage MOGCT cases may be possible candidates for NACT; however, suitable candidates are still unknown.

Except for this study [9], no other studies have proposed a potential candidate for NACT. Although the ideal number of NACT cycles is unclear, most studies performed one to three cycles of NACT, and a Chinese study found that the tumor size tended to decrease to a lesser degree after each successive cycle of NACT (15% versus 3.3% versus 1.0%) [9]. Based on the limited evidence, a large MOGCT tumor size may be a possible candidate for NACT, and the appropriate number of NACT cycles may range between 1 and 3 (less is better). However, further studies are required to validate these findings.

#### 4.3.3. Additional Outcomes

In the RCOG Green Top guidelines, NACT is recommended for advanced disease to help preserve fertility [50]. In the present study, the rate of uterine preservation (number of uterine preservations/total number of cases) was approximately 70% in women who underwent NACT; however, the success rate of uterine preservation (number of uterine preservations/women who desired uterine preservation) was unavailable from the currently retrieved evidence. Overall, it remains unknown whether NACT helps with uterine preservation in women with advanced MOGCT. We believe that NACT may help uterine preservation, as the response rate of NACT is approximately 95%, and reducing the complexity of IDS may increase the rate of fertility-sparing surgeries.

## 5. Conclusions

### 5.1. Implications for Practice

The use of NACT for EOC is aimed at decreasing surgical morbidity during IDS with non-inferior oncologic outcomes compared to PDS. Similar NACT effects are expected in the clinical management of patients with MOGCT. Nevertheless, women with MOGCT are usually young, and an improvement in the fertility-sparing rate is expected in women with advanced MOGCT.

### 5.2. Implications for Clinical Research

Since the standard treatment has already shown relatively good outcomes for MOGCT, it is difficult to identify NACT candidates for MOGCT. While the data were from only one study, DFS was superior in the NACT group than in the PDS group in women with large MOGCT tumors. Future studies focused on identifying patients with MOGCT who are promising candidates for NACT are warranted.

## Figures and Tables

**Figure 1 cancers-15-04470-f001:**
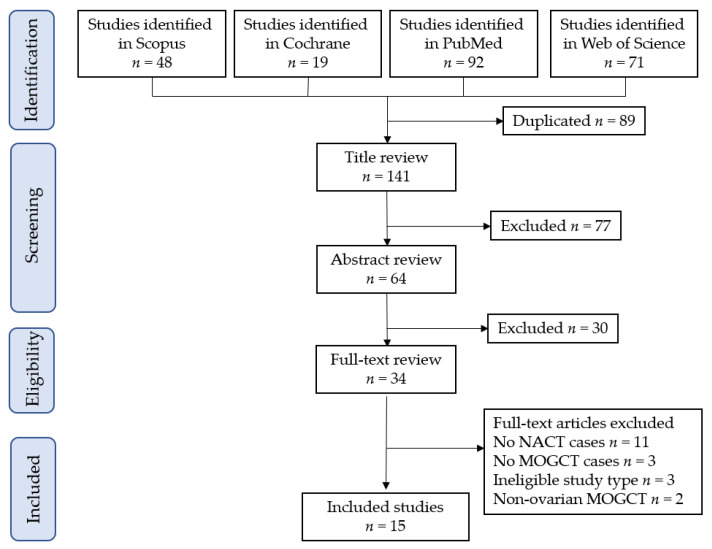
Study selection scheme of the current systematic review.

**Figure 2 cancers-15-04470-f002:**
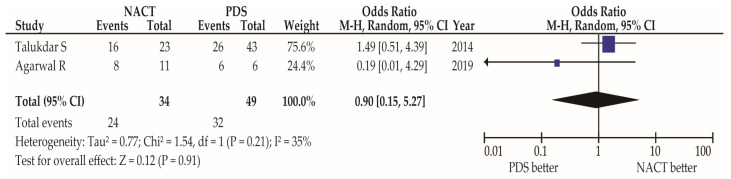
Complete response rate between groups of NACT and PDS in women with MOGCT [30,34].

**Table 1 cancers-15-04470-t001:** The rate of use, regimen, and cycles of NACT use.

Author	Year	Total	Age ^‡^	II-IV	NACT	Rate (all)	Rate (adv)	Regimen ^†^	Cycles
Agrawal A [28]	2023	31	19 (11–48)	--	2	6.5%	--	TC, BEP (1,1)	4
Newton C [29]	2019	138	23.6 (8–76)	49 (35.5%)	16	11.6%	32.7%	--	--
Agarwal R [30]	2019	48	20.5 (9–45)	17 (35.4%)	11	22.9%	64.7%	--	3–4
Divya S [31]	2019	10	20.5 (13–21)	--	2	25%	--	BEC (2)	--
Lakshmanan M [32]	2018	39	22 (11–65)	33 (84.6%)	27	69.2%	81.8%	BEP (27)	--
Zhang GY [33]	2018	58	--	--	18	31.0%	--	BEP (18)	1–3
Lu Y [9]	2014	53	--	53 (100%)	21	39.6%	39.6%	Cis-based	1–3
Talukdar S [34]	2014	66	--	66 (100%)	23	34.8%	34.8%	BEP (23)	4
Bafna UD [36]	2001	33	17 (4–32)	--	4	12.1%	--	BEP (4)	2–3
Baranzelli MC [37]	2000	49	--	48 (98.0%)	12	24.5%	25.0%	Pt-based	--

Some of the values were estimated by the authors. The number of cases is indicated in parentheses. Abbreviations: II–IV, number of women with stages II–IV; rate (all), the rate of NACT use in women with MOGCT; rate (adv), the rate of NACT use in women with advanced MOGCT (stages II–IV); NACT, the number of women who underwent neoadjuvant chemotherapy; Cis, Cisplatin; Pt, platinum drugs; TC, paclitaxel + carboplatin; BEP, bleomycin + etoposide + cisplatin; BEC, bleomycin + etoposide + carboplatin; and --, not applicable. ^†^ NACT regimen. ^‡^ median age with a range for the whole cohort.

**Table 2 cancers-15-04470-t002:** Response rate to therapy, survival outcomes, and resumed menstruation rate (NACT versus PDS).

Response Rate (Stage III–IV)			NACT				PDS		
	Year	*N*	RR	CR	PR	*N*	RR	CR	PR
Agarwal R [30]	2019	11	>72.7%	8	--	6	100%	6	--
Talukdar S [34]	2014	23	91.30%	16	5	43	76.70%	26	7
Survival rate	Year	*N*	NACT			*N*	PDS		
Agarwal R [30]	2019	11	5-yr OS: 100%			6	5-yr OS: 100%		
			5-yr DFS: 100%				5-yr DFS: 100%		
Lu Y [9]	2014	21	5-yr OS: 95.3% ^$^			32	5-yr OS: 96.9% ^$^		
			5-yr DFS: 95.3% ^$^				5-yr DFS: 96.9% ^$^		
Talukdar S [34]	2014	23	5-yr OS: 87.0%			43	5-yr OS: 70.0%		
			5-yr DFS: 87.0%				5-yr DFS: 70.0%		
Tumor size > 20 cm	Year	*N*	NACT			*N*	PDS		
Lu Y [9]	2014	16	3-yr DFS: 80.0% ^$^			4	3-yr DFS: 25.0% ^$^		
Recurrence rate	Year	*N*	NACT			*N*	PDS		
Lakshmanan M [32]	2018	27	7 (25.9%)			11	1 (9.1%) ^$^		
IDS	Year	*N*	NACT			*N*	PDS		
Lu Y [9]	2014	# similar surgical outcomes except for intraoperative blood loss.							
Resumed menstruation	Year	*N*	NACT			*N*	PDS		
Agarwal R [30]	2019	11	11 (100%)			4	4 (100%)		
Talukdar S [34]	2014	18	18 (100%)			30	17 (56.6%)		
AE of chemo (G3/4)	Year	*N*	NACT			*N*	PDS		
Talukdar S [34]	2014								
Overall		## similar adverse effects between the two groups.							
Leukopenia *		23	6 (26.8%)			43	11 (25.5%)		
Thrombocytopenia *		23	1 (4.3%)			43	6 (14.0%)		
Pulmonary toxicity *		23	0 (0%) ^$^			43	0 (0%) ^$^		

Some of the values were estimated by the authors. Abbreviations: MOGCT, malignant ovarian germ cell tumor; NACT, neoadjuvant chemotherapy; PDS, primary debulking surgery; IDS, interval debulking surgery; *N*, total number of included cases; OS, overall survival; DFS, disease-free survival; RR, response rate (CR + PR); CR, complete response; PR, partial response; PD, progression of disease; AE, adverse events; chemo, chemotherapy; vs., versus; G, grade. # The rates of transfusion, operation time, mean hospital stay (days), and postoperative complications were similar between the two groups. ^$^ estimated by the authors; * statistically not significant (analyzed by the authors using Fisher’s exact test); ## NACT vs. PDS: anemia (2/23 [8.7%] vs. 4/43 [9.3%]), nausea/vomiting (1/23 [4.3%] vs. 4/43 [9.3%]), mucositis (1/23 [4.3%] vs. 3/43 [7.0%]), diarrhea (1/23 [4.3%] vs. 1/43 [2.3%]), renal (0/23 [0%] vs. 0/43 [0%]), fever (0/23 [0%] vs. 2/43 [4.6%]), alopecia (2/23 [8.7%] vs. 4/43 [9.3%]).

**Table 3 cancers-15-04470-t003:** Response rate to NACT for MOGCT.

Author	Year	Total	NACT	RR	CR	PR	PD
Agarwal R [30]	2019	48	11	>72.7%	8	--	0 ^$^
Lakshmanan M [32]	2018	39	27	100%	6	21	0
Zhang GY [33]	2018	58	18	94.4%	0	17	0
Lu Y [9]	2014	53	21	>14.3%	3 ^#^	--	0 ^$^
Talukdar S [34]	2014	66	23	91.3%	16	5	1
Bafna UD [36]	2001	33	4	100%	0	4 ^$^	0 ^$^

Some of the values were estimated by the authors. MOGCT, malignant ovarian germ cell tumors; Total, total number of included cases; NACT, neoadjuvant chemotherapy; RR, response rate (CR + PR); CR, complete response; PR, partial response; PD, progression of disease. ^$^ estimated by the authors. ^#^ Pathological CR.

**Table 4 cancers-15-04470-t004:** Survival outcomes of NACT (non-comparator studies).

Author	Year	NACT	Hist ^#^	OS	DFS
Agarwal R [30]	2019	11	Dys, MG, IT, YST	100%	100%
Zhang GY [33]	2018	18	YST	94.4%	94.4%
Lu Y [9]	2014	21	YST	95.3% ^$^	95.3% ^$^
Talukdar S [34]	2014	23	Dys, YST, MG	87.0%	87.0%

Some of the values were estimated by the authors. MOGCT, malignant ovarian germ cell tumor; Hist, histological subtype; NACT, neoadjuvant chemotherapy; PDS, primary debulking surgery; IT, immature teratoma; YST, yolk sac tumor; IDS, interval debulking surgery; OS, overall survival; DFS, disease-free survival; Dys, dysgerminoma; MG, mixed germ cell tumor; YST, yolk sac tumor. ^#^ NACT cases; ^$^, estimated by the authors.

**Table 5 cancers-15-04470-t005:** Rates of hysterectomy, resumed menstruation, and pregnancy.

Author	Year	Total	Stage	II-IV	NACT	Hyst	Mens	Preg
Agrawal A [28]	2023	31	I-IV	--	2	1/2	1/1	--
Agarwal R [30]	2019	48	I-IV	17	11	0/11	11/11	--
Lakshmanan M [32]	2018	39	I-III	33	27	22/27	--	--
Zhang GY [33]	2018	58	III-IV	--	18	1/18	--	6/6
Talukdar S [34]	2014	66	III-IV	66	23	3/21	18/18	10/10

Some of the values were estimated by the authors. Abbreviations: NACT, the number of women who underwent neoadjuvant chemotherapy; II–IV, number of women with stages II–IV; hyst, rate of hysterectomy; mens, rate of resumed menstruation; preg, rate of pregnancy; --, not applicable.

## Data Availability

All published studies cited and used in this report are published in the literature and are publicly available.

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
