# Peer review of "Systematic Review of the Survival Outcomes of Neoadjuvant Chemotherapy in Women with Malignant Ovarian Germ Cell Tumors"

_cancers, 2023, doi:10.3390/cancers15184470_

Round 1
Reviewer 1 Report
With pleasure, I read the paper titled “Survival outcomes of neoadjuvant chemotherapy in women with malignant ovarian germ cell tumors: systematic review and meta-analysis”. The topic is clinically relevant to practice and of importance to the readers of the Cancers journal. Overall, the manuscript reads well and has good flow of ideas, up-to-date citations, and proper summary of data using tables and figures. A major strength of the article is being the first-ever to examine the role of NACT in patients with MOGCTs. The rationale of the study is clearly presented in the introduction section. The research is well-articulated to encourage more research in the field. Despite the data suggested that there might be no substantial difference between NACT and PDS, nonetheless, there could be a trend for potential benefit in favor of NACT. It remains a pivotal area of research to identify MOGCT patients are likely to benefit the most from administration of NACT. The research has some key and unavoidable limitations, all of which have been explicitly acknowledge, which include the small number of included studies, low-quality study designs (retrospective case series and case reports), confounding factors, among others. The conclusion is well-written. The authors did a wonderful work handling the extremely limited available data. This manuscript is well-written and is very likely to be cited extensively in the future. I have the following questions:
Title. The paper is mostly a systematic review without an extensive meta-analysis component. Hence, I recommend omitting the word “meta-analysis” from the title.
Abstract. Please indicate the total number of studies inclusive of original and case reports. The data from comparative studies (NACT vs PDS) are important and should be reported with their ORs/HRs and 95% CIs.
Introduction. Please highlight the significance of your research by indicating your research is the first-ever systematic review and/or meta-analysis on the topic.
Methods. How was the quality of case reports assessed? For assessment of between-study heterogeneity, have you also considered the p-value of the Cochran’s Q test (i.e., p<0.1)? Have the authors examined (i) the gray literature or (ii) the reference lists of the included studies for other potential citations/studies that could have been mistakenly missed during literature screening?
Results. For all meta-analysis results, please summarize the data in forest plots so the readers can appreciate better the number of pooled studies, heterogeneity of studies, p-values of statistical significance, and the model used for analysis (random- of fixed-effects).
Discussion. Please add one more limitation which is that most studies (n=8) originated from India, hence negatively impacting the generalizability of the data. Does the definition of advanced stage disease include stage II-IV or III-IV?
Author Response
We would like to thank the Editor and the Reviewers for their insightful comments. The following are our point-by-point responses to the comments and explanations of the revisions made to the manuscript. The revisions are indicated using the “track changes” function of Microsoft Word in the manuscript file; the line numbers of revisions corresponding to each comment are indicated below.
Reviewer #1
With pleasure, I read the paper titled “Survival outcomes of neoadjuvant chemotherapy in women with malignant ovarian germ cell tumors: systematic review and meta-analysis”. The topic is clinically relevant to practice and of importance to the readers of the Cancers journal. Overall, the manuscript reads well and has good flow of ideas, up-to-date citations, and proper summary of data using tables and figures. A major strength of the article is being the first-ever to examine the role of NACT in patients with MOGCTs. The rationale of the study is clearly presented in the introduction section. The research is well-articulated to encourage more research in the field. Despite the data suggested that there might be no substantial difference between NACT and PDS, nonetheless, there could be a trend for potential benefit in favor of NACT. It remains a pivotal area of research to identify MOGCT patients are likely to benefit the most from administration of NACT. The research has some key and unavoidable limitations, all of which have been explicitly acknowledge, which include the small number of included studies, low-quality study designs (retrospective case series and case reports), confounding factors, among others. The conclusion is well-written. The authors did a wonderful work handling the extremely limited available data. This manuscript is well-written and is very likely to be cited extensively in the future. I have the following questions:
Reply:
Thank you for your positive comments and for your constructive critique to improve the manuscript. We have made every effort to address the issues raised and to respond to all comments. Please find below a detailed, point-by-point response to the reviewer's comments. We hope that our revisions will meet your expectations.
Reviewer #1, comment #1
Title. The paper is mostly a systematic review without an extensive meta-analysis component. Hence, I recommend omitting the word “meta-analysis” from the title.
Reply: Title
Thank you for your insightful comments. Accordingly, we have removed “meta-analysis” from the title.
Reviewer #1, comment #2
Abstract. Please indicate the total number of studies inclusive of original and case reports. The data from comparative studies (NACT vs PDS) are important and should be reported with their ORs/HRs and 95% CIs.
Reply: Abstract, Discussion, Line 406
We appreciate this comment. We have added the total number of articles identified in the systematic literature search. We have also added the odds ratios for response rates by comparing the neoadjuvant group and the primary debulking surgery group. Because no studies have calculated the hazard ratio for overall survival and disease-free survival, we were not able to calculate a pooled hazard ratio. We believe this point is a weakness of the current study; thus, we have added this point as a limitation of this study.
Reviewer #1, comment #3
Introduction. Please highlight the significance of your research by indicating your research is the first-ever systematic review and/or meta-analysis on the topic.
Reply: Introduction, Line 77
Thank you for your helpful suggestion. We have highlighted the significance of the current study by indicating that, to the best of our knowledge, this is the first systematic review regarding neoadjuvant chemotherapy (NACT) for women with malignant ovarian germ cell tumors (MOGCTs).
Reviewer #1, comment #4
Methods. How was the quality of case reports assessed?
Reply: Results, Lines 191–192, lines 197–198
Because the analysis for case reports was not performed, the risk of bias was not assessed for case reports. We have clarified this point in lines 197–198.
Reviewer #1, comment #5
For assessment of between-study heterogeneity, have you also considered the p-value of the Cochran’s Q test (i.e., p<0.1)?
Reply: Materials and Methods, Lines 151–155
Thank you for your helpful comment. Owing to the limited number of available studies, only one forest plot was generated. A pooled odds ratio for the response rate of chemotherapy was calculated by combining results from two studies. We used I2 statistics to evaluate the heterogeneity of the included studies and have clarified this in the Methods section.
Reviewer #1, comment #6
Have the authors examined (i) the gray literature or (ii) the reference lists of the included studies for other potential citations/studies that could have been mistakenly missed during literature screening?
Reply: Materials and Methods, Lines 124–126, lines 172–174
Thank you for your thoughtful suggestion. In this study, we did not include the gray pieces of literature and have clarified this point in the exclusion criteria of the current study. According to the registered protocol of the current systematic review, we did not add reference lists of other potential studies. Nevertheless, we agree with the importance of the accuracy of the literature search. We checked the reference lists of all eligible studies and confirmed that potential citations/studies were not missed during literature screening.
Reviewer #1, comment #7
Results. For all meta-analysis results, please summarize the data in forest plots so the readers can appreciate better the number of pooled studies, heterogeneity of studies, p-values of statistical significance, and the model used for analysis (random- of fixed-effects).
Reply: Discussion, Figure 2, line 405
We appreciate the reviewer’s helpful suggestion. A forest plot was created for the response rate of NACT, including two studies. With regard to the rate of resumed menstruation, all women resumed menstruation in Agarwal R’s study, and an odds ratio was not calculated. Thus, a pooled odds ratio could not be calculated regarding the rate of resumed menstruation. Survival outcomes could not be combined because all studies did not calculate hazard ratios comparing the neoadjuvant chemotherapy and primary debulking surgery groups. We think this point is a weakness of this study and have added it as a limitation.
Reviewer #1, comment #8
Discussion. Please add one more limitation which is that most studies (n=8) originated from India, hence negatively impacting the generalizability of the data.
Reply: Discussion, Lines 427–428
We appreciate the reviewer’s suggestion. Accordingly, we have added the negative impact on the generalizability of the results of the current study due to studies reported from limited countries or geographic regions.
Reviewer #1, comment #9
Does the definition of advanced stage disease include stage II-IV or III-IV?
Reply: Discussion, Lines 445–453
We have clarified the definition of the “advanced stage” referred to in national guidelines.

Reviewer 2 Report
This paper is a systematic review and meta-analysis assessing the possible role of neoadjuvant chemotherapy (NACT) for malignant ovarian germ cell tumors (MOGCT) focusing on oncologic outcomes in terms of response rate, overall survival, disease-free survival, and recurrence rate. Adverse events rate from chemotherapy and surgery, and fertility outcome (rates of hysterectomy, resumed menstruation, and pregnancy) were also taken into account.
The paper is well written and the English language is appropriate and understandable.
The clinical topics are extremely interesting due to the rarity of MOGCT typically occurring in young women, and because the effect of NACT on women with MOGCT is under-evaluated (most of the patients are at an early stage). To date available data are limited and level I evidence is lacking.
This paper shows that NACT is feasible for advanced germ cell tumors with comparable oncologic and fertility outcomes between the NACT and primary debulking surgery groups. It is well known that MOGCTs are characterized by high chemosensitivity.
Of course, the results of the current systematic review are the first to appear in the literature focusing on the outcomes of women with MOGCT who underwent NACT. However, they require careful interpretation, and further research is warranted to demonstrate the efficacy of NACT for advanced MOGCT.
The limitations and bias of this review are correctly reported by the Authors. All identified studies were retrospective, with a limited number of included cases, and without a standard policy regarding the decision to use NACT. No studies have assessed its central pathology and few data were available on histological subtypes of MOGCT and chemotherapy regimens.
Author Response
We sincerely appreciate the reviewer’s positive comments. We trust that the revised manuscript will now be suitable for publication in Cancers.
Reviewer 3 Report
The review manuscript “ Survival outcomes of neoadjuvant chemotherapy in women with malignant ovarian germ cell tumors: Systematic review and meta-analysis” by Hitomi Sakaguchi-Mukaida and co-authors to examine the effects of neoadjuvant chemotherapy (NACT) in women with malignant ovarian germ cell tumors (MOGCT) by conducting a systematic review of four public search engines. Among the 10 eligible original articles, NACT was used in approximately 40% of the advanced MOGCT cases. Most women were treated with a Bleomycin, Etoposide, and Cisplatin regimen, and 1-3 cycles were used in most studies. In six studies, comprising 104 NACT cases, the cumulative rate of complete response to chemotherapy was 31.7%, and the response rate was 95.8%. The uterine preservation rate was approximately 70%. Four studies comparing NACT and primary debulking surgery showed similar values for overall survival, disease-free survival, recurrence rate, and adverse events rate from chemotherapy between the groups. In conclusion, NACT may be considered for the management of MOGCT; however, possible candidates for NACT use and ideal number of NACT cycles remain unknown. Further studies are warranted to validate the efficacy of NACT in advanced MOGCT patients. However, some concerns that must be taken into account before the work can be reconsidered for publication.
Comment
1. Figure 1: Why did author exclude patients (the excluded n=77 excluded n=30)? The excluded reasons should be added.
2. Table 1: The percentage (%) should be added in table 1.
3. Lane 296: “the response rate to NACT (CR + PR) was 95.8% (69/72 women)” How to calculate?
4. Lane 314-322: It is hard to follow the percentage. How to calculate?
Extensive editing of English language required
Author Response
We would like to thank the Editor and the Reviewers for their insightful comments. The following are our point-by-point responses to the comments and explanations of the revisions made to the manuscript. The revisions are indicated using the “track changes” function of Microsoft Word in the manuscript file; the line numbers of revisions corresponding to each comment are indicated below.
Reviewer #3
The review manuscript “ Survival outcomes of neoadjuvant chemotherapy in women with malignant ovarian germ cell tumors: Systematic review and meta-analysis” by Hitomi Sakaguchi-Mukaida and co-authors to examine the effects of neoadjuvant chemotherapy (NACT) in women with malignant ovarian germ cell tumors (MOGCT) by conducting a systematic review of four public search engines. Among the 10 eligible original articles, NACT was used in approximately 40% of the advanced MOGCT cases. Most women were treated with a Bleomycin, Etoposide, and Cisplatin regimen, and 1-3 cycles were used in most studies. In six studies, comprising 104 NACT cases, the cumulative rate of complete response to chemotherapy was 31.7%, and the response rate was 95.8%. The uterine preservation rate was approximately 70%. Four studies comparing NACT and primary debulking surgery showed similar values for overall survival, disease-free survival, recurrence rate, and adverse events rate from chemotherapy between the groups. In conclusion, NACT may be considered for the management of MOGCT; however, possible candidates for NACT use and ideal number of NACT cycles remain unknown. Further studies are warranted to validate the efficacy of NACT in advanced MOGCT patients. However, some concerns that must be taken into account before the work can be reconsidered for publication.
Reviewer #3, comment #1
Comment
- Figure 1: Why did author exclude patients (the excluded n=77 excluded n=30)? The excluded reasons should be added.
Reply: Figure 1
Thank you for your helpful comment. We have added text clarifying study selection as follows: “After excluding duplicated studies, 141 potentially eligible studies were screened. First, 77 studies were excluded after a title review. Second, 30 studies were excluded during an abstract review. Third, 19 studies were excluded after a full-text review; reasons for exclusion are presented in Figure 1.”
Reviewer #3, comment #2
- Table 1: The percentage (%) should be added in table 1.
Reply: Table 1, Lines 206–208
Thank you for your helpful comment. We have added the percentage of women with advanced-stage MOGCT in Table 1 and in the main text.
Reviewer #3, comment #3
- Lane 296: “the response rate to NACT (CR + PR) was 95.8% (69/72 women)” How to calculate?
Reply: Results, Lines 334–339
We apologize for the inadequate description of the response rate to NACT. We have revised the sentences, adding an explanation of the calculation of the response rate to NACT.
Reviewer #3, comment #4
- Lane 314-322: It is hard to follow the percentage. How to calculate?
Reply: Results, Lines 357–368
Thank you for your helpful comment. Accordingly, we have added an explanation to improve readability.

Reviewer 4 Report
This study aimed to examine the effects of NACT in women with MOGCT by conducting a systematic review. This manuscript was well written and organized. I only have some comments to the authors.
1. Please summarize the median age (IQR) in the Table 1.
2. Please compare the adverse events in more detail (e.g. hematological toxicity and pulmonary toxicity due to bleomycin). This information would be helpful to evaluate the benefit of neoadjuvant chemotherapy.
3. The evaluation of response of NACT is important. It is of interest how and when would be better to evaluate the response of NACT, and decide the timing of surgery. In patients with stable disease or poor response after NACT, would stable disease or poor response a predictor of poor outcomes in these patients? Please comment this point in the Discussion section (4.3.1. Primary outcomes: response rate).
Dear Editor,
Thank you for invitation for reviewing this manuscript. Overall, this manuscript is well organized, and bears clinical relevance. I only have some comments to the authors. Thank you very much!
Author Response
We would like to thank the Editor and the Reviewers for their insightful comments. The following are our point-by-point responses to the comments and explanations of the revisions made to the manuscript. The revisions are indicated using the “track changes” function of Microsoft Word in the manuscript file; the line numbers of revisions corresponding to each comment are indicated below.
Reviewer #4
This study aimed to examine the effects of NACT in women with MOGCT by conducting a systematic review. This manuscript was well written and organized. I only have some comments to the authors.
Reviewer #4, comment #1
- Please summarize the median age (IQR) in the Table 1.
Reply: Table 1, lines 206–208
Thank you for your helpful comment. Accordingly, we have added the median age range for the whole cohort. Because the interquartile range was not available in most studies, we have indicated the lower and upper ranges of age.
Reviewer #4, comment #2
- Please compare the adverse events in more detail (e.g. hematological toxicity and pulmonary toxicity due to bleomycin). This information would be helpful to evaluate the benefit of neoadjuvant chemotherapy.
Reply: Results, Lines 291–294, Table 2
We appreciate your comment regarding the adverse effects of chemotherapy. We agree with the importance of the adverse effects of chemotherapy and have added a description of hematological toxicity and pulmonary toxicity due to bleomycin.
Reviewer #4, comment #3
- The evaluation of response of NACT is important. It is of interest how and when would be better to evaluate the response of NACT, and decide the timing of surgery. In patients with stable disease or poor response after NACT, would stable disease or poor response a predictor of poor outcomes in these patients? Please comment this point in the Discussion section (4.3.1. Primary outcomes: response rate).
Reply: Discussion, Lines 473–498
Thank you for your insightful comment. Accordingly, we have added the results of the literature review regarding the evaluation of NACT, including the method, timing, and decision of surgery. We have also added outcomes for poor responders to NACT.

Round 2
Reviewer 3 Report
The revised manuscript “Systematic review of the survival outcomes of neoadjuvant chemotherapy in women with malignant ovarian germ cell tumors” have adequately addressed my previous concerns and the paper is now acceptable for publication.